# Fracture Behavior of Steel Slag Powder-Cement-Based Concrete with Different Steel-Slag-Powder Replacement Ratios

**DOI:** 10.3390/ma15062243

**Published:** 2022-03-18

**Authors:** Ke-Xian Zhuo, Guo-Tao Liu, Xue-Wei Lan, Dong-Ping Zheng, Si-Quan Wu, Pei-Zong Wu, Yong-Chang Guo, Jia-Xiang Lin

**Affiliations:** 1School of Civil and Transportation Engineering, Guangdong University of Technology, Guangzhou 510006, China; kexzhuo@163.com (K.-X.Z.); cosonz@163.com (D.-P.Z.); a623633772@163.com (S.-Q.W.); a623633772@126.com (P.-Z.W.); guoyc@gdut.edu.cn (Y.-C.G.); 2Guangdong GW Metal Industry Group Co., Ltd., Guangzhou 510030, China; gwjsim@gwjscyjt.com; 3Guangzhou Zengcheng Zhengyuan Construction Engineering Testing Center Co., Ltd., Guangzhou 511300, China; zyjc06@163.com; 4Guangzhou Hongchang Construction Technology Co., Ltd., Guangzhou 510006, China

**Keywords:** steel-slag powder, replacement ratio, fracture performance, concrete

## Abstract

The influence of different replacement ratios of steel-slag powder as cement-replacement material on the fracture performance of concrete is studied in this paper. A three-point bending fracture test is carried out on slag powder-cement-based concrete (SPC)-notched beams with steel-slag powder as cementitious materials, partially replacing cement (0%, 5%, 10%, 15%, and 20%). Load-deflection curves and load-crack-opening displacement curves of SPC-notched beams with five different replacement ratios of steel-slag powder were obtained. The effects of different steel-slag-powder replacement ratios on the fracture properties (fracture energy, fracture toughness, and double-K fracture parameters) of the SPC were analyzed and discussed. The results showed that the incorporation of appropriate steel-slag powder can affect the fracture performance of SPC. Compared with concrete without steel-slag powder, adding appropriate steel-slag powder can effectively improve the bond performance between aggregate and matrix because the steel-slag powder contains hydration activity substances such as calcium oxide and aluminium trioxide. The fracture energy and fracture toughness of SPC increased and then decreased with the increase in steel-slag-powder replacement ratios, and the SPC concrete showed best fracture performance with a 5% steel slag powder replacement ratio. Its fracture energy increases by 13.63% and fracture toughness increases by 53.22% compared with NC.

## 1. Introduction

With the continuous development of civil-engineering technology, many new building materials are emerging, such as high-strength aluminum alloy, fiber concrete, sea-sand concrete, FRP materials, recycled concrete, steel-slag concrete, and so on [1,2,3,4]. Among them, steel-slag concrete is an environmental protection material [1]. Steel slag is a by-product of the steelmaking industry, accounting for emissions of about 10% to 20% in steel production [5,6]. China is a major industrial country with yearly production of steel consumption about 50% of the world’s, resulting in an annual production of steel slag of approximately 10 million tons [7,8]. In recent years, how to turn waste steel slag into valuables has become a hotspot. Steel slag contains tricalcium silicate and dicalcium silicate mineral phases with similar composition to cement, as well as a slight amount of free calcium oxide and metallic iron [4,9]. Some researchers have proposed the use of steel slag as a coarse and fine aggregate for concrete, or finely ground into powder as cement admixture, as an effective way to improve the comprehensive utilization of steel slag [10,11,12,13]. Many studies have shown that when steel-slag powder is used as a cementitious material instead of cement, the steel-slag powder has a cementing activity that occurs under the mixture system with continuous hydration, which not only effectively reduces the amount of cement, but also improves the mechanical properties of concrete [12,14,15]. However, it is not that the higher steel-slag-powder replacement ratio is better. It was found that when the steel-slag-powder replacement ratio is 10%, the compressive strength of slag powder-cement-based concrete (SPC) is better than that of ordinary concrete (NC) and has good compatibility [14,16]. The steel-slag powder makes the concrete denser and effectively improves the impermeability of concrete, but the compressive strength gradually decreases with the increase in the steel-slag-powder replacement ratio. In addition, steel slag significantly improves the compressive strength, water absorption, and durability of concrete when used either as a coarse and fine aggregate in concrete [17,18,19,20]. Steel-slag concrete has promising advantages, not only for improving the workability of concrete, but also for being widely used in seismic-resistant buildings [21,22,23]. At present, several researchers have carried out various mechanical-property studies on steel-slag concrete [6,24,25,26]. The effect of expanding the range of steel-slag admixture on the compressive strength of normal and high-strength concrete has been studied [26]. The results showed that the appropriate amount of steel-slag incorporation not only improved the compressive strength of concrete, but also had a positive effect on the energy-dissipation capacity, especially more pronounced in normal-strength concrete. Palod, R. et al. [27] studied the properties of concrete with steel-slag powder as a substitute for cement with 20% replacement, which showed a promising and excellent performance on compressive strength, splitting-tensile strength, early-shrinkage properties, and resistivity. When steel slag was used as a partial replacement of cement in the UHPC mix [28], although the compressive strength decreased slightly, the self-shrinkage performance was improved with the increase in slag content, which was beneficial to the utilization of slag in UHPC. Steel-slag-concrete mixes have shown durability performance and mechanical stability as good as sea-sand concrete in acid and alkaline, freeze-thaw, and marine environments [29,30,31]. Saxena, S. et al. [18] used steel-slag aggregate to replace natural coarse aggregate at 15%, 25%, 50%, 75%, and 100% to study the compressive strength and flexural strength of concrete. The results indicated that the replacement of 50% basalt aggregate by steel-slag aggregate increased the compressive strength, flexural strength, and modulus of elasticity of concrete by 33% and 9.8%, respectively. Steel-slag concrete also exhibits better strain-ratio effects in terms of dynamics. Guo, Y.C. et al. [6] investigated the axial-impact compression properties of steel-slag-sand concrete (SSC) using 0%, 10%, 20%, 30%, and 40% steel-slag fine aggregates, respectively. The results of the study showed that the addition of steel slag as a fine aggregate can improve the dynamic properties of steel-slag concrete. In a further study, Rooholamini, H. et al. [32] examined the effect of partial and total replacement of natural aggregates by steel slag on the mechanical and fracture properties of compacted concrete, which were improved due to the high angularity and roughness of the coarse steel slag. It can be seen that the field of steel-slag-concrete research is mainly focused on use of steel-slag coarse and fine aggregates in concrete mixes. Steel slag has good promotion and environmental significance as a cement material with active material, as a partial replacement of cement in the preparation of concrete. At present, extensive research has been devoted to the properties of slag powder-cement-based concrete, and the results indicate that SPC is advantageous in terms of the mechanical strength and durability of the hardened mortars and concretes [33,34,35]. The addition of 10% steel-slag powder in SPC resulted in 6.5% and 11.9% higher compressive and tensile strength, respectively. When the dosage of the steel-slag powder was more than 10%, the strength clearly decreased [36]. In contrast to the incorporated steel slag in the15–30 μm and 30–45 μm ranges, the steel-slag particles with sizes of less than 15 μm and 45–80 μm were advantageous for the 7- and 28-day compressive-strength development of the concrete, especially at a low replacement ratio of up to 10% [37]. In conclusion, the mechanical properties of SPC are mainly focused on axial compressive and tensile behaviors, while the fracture properties of steel-slag-powder concrete are not clear.

Fracture-mechanics theory has been developed for decades. In 1961, Kaplan, M.F. et al. [38] applied the theory of fracture mechanics to concrete problems for the first time and conducted tests on the fracture toughness of concrete. Hillerborg, A. et al. [39] established a fracture model suitable for materials on concrete and deduced many valuable research results. Many studies have investigated the feasibility and effectiveness of fracture-mechanics theory in the study of different concrete mixes to further advance the research process of concrete fracture properties [40,41,42]. Since studies on fracture-mechanical properties of steel-slag powder-concrete have not been reported in the literature, and fracture properties are important parameters for structural-safety assessment and crack control, it is necessary to investigate the fracture properties of SPC. In this paper, based on the current developments in concrete fracture-mechanics theory, beam specimens of five groups of SPC mix, with different steel-slag-powder replacement ratios, were subjected to three-point bending fracture tests to explore the effect of substitution of steel-slag powder for cement, on the fracture properties of concrete, and to provide a basis for promoting the use of steel slag as an environmentally compatible building material.

## 2. Experimental Design

### 2.1. Test Materials

The composition of the specimens is mainly cement, steel-slag powder, river sand, granite crushed stone, water-reducing agent, and water. Among them, the cement is P.0 42.5 R ordinary silicate cement with the apparent density of 3100.0 kg/m^3^. The apparent density of steel-slag powder is 2941.2 kg/m^3^. The 9.7% particle size distribution is 0.09~0.30 mm and the 90.3% particle size distribution is 0.30~0.60 mm, produced by Lingshou Haotai mine processing plant in Hebei Province of China. The chemical compositions of the cement and steel-slag powder used in this study were determined by X-ray fluorescence method are as shown in Table 1. The crystalline phase of the steel slag was determined by powder X-ray diffraction (XRD) as shown in Figure 1. As can be seen from Table 1, the steel-slag powder has high CaO and Al_2_O_3_ content and low SiO_2_ content, and has high activity. From Figure 1, it can be seen that the crystalline phase of steel-slag powder has more gel components, such as C_2_S and C_3_S. The fineness modulus of 2.52 of river sand, with an apparent density of 2678.6 kg/m^3^ and with a water absorption ratio of 0.8%, are utilized. The natural coarse aggregate is granite crushed stone with apparent density of 2740.0 kg/m^3^, water absorption ratio of 0.75%, and continuous gradation of grain size 4.75~25 mm. The additive, a reddish-brown liquid, is QL-pc5 polycarboxylic acid high-efficiency water-reducing agent with 20% water-reduction ratio and air content 1–5%, which was provided by the manufacturer. The amount of superplasticizer used in the concrete mixtures was set at 0.5% by weight of cement to meet the slump of 150 mm, which was based on the slump tests in accordance with Chinese standard [43].

### 2.2. Specimens and Mix Proportions

In order to explore the effect of different replacement ratios of steel-slag powder on the fracture properties of the SPC, beam specimens of a total of five groups of specimens with 0%, 5%, 10%, 15%, and 20% replacement ratios of steel-slag powder were tested, respectively. Mix proportions are shown in Table 2. According to the recommendation of the International Commission on Fracture Mechanics, the three-point bending experimental method is used in this paper to determine the fracture properties of slag-concrete beams [44] with 15 specimens with dimensions of 150 mm × 150 mm × 550 mm (height × width × length), span of 450 mm, prefabricated crack depth of 45 mm, and thickness of 2 mm, as shown in Figure 2. Before the test, the specimens were placed at room temperature for 24 h and sealed with plastic film to maintain wet conditions, and the specimens were stored in a cool place and cured for 28 days by sprinkling water every day after demolding.

### 2.3. Experimental Setup

According to the recommendation of the International Commission on Fracture Mechanics, a three-point bending fracture test is carried out in this paper to investigate the fracture properties of SPC-notched beams [44,45,46]. The experimental setup is shown in Figure 2. A universal material-testing machine was adopted to perform the three-point bending fracture test. Displacement-loading mode was selected, with a loading ratio of 0.1 mm/min [47]. Mid-span load, mid-span deflection, and crack-tension displacement were collected. The loads in the span were measured by the load cell of the testing machine. The mid-span deflection measurement was measured by a displacement gauge (LVDT) with an accuracy of 0.01 mm. The crack mouth-opening displacement-measurement (CMOD) instrument is a high-precision clip-on extensometer. The load and deformation data were collected simultaneously by the high-speed static strain-acquisition system TDS-530.

## 3. Results and Analysis

### 3.1. Fracture Mode

The fracture morphology of the specimens with different steel-slag-powder replacement ratios is shown in Figure 3. Three kinds of fracture morphology of coarse-aggregate fracture, cement-matrix fracture, and coarse-aggregate surface were captured, as can be seen from Figure 3. Different fracture-morphology distributions are observed in fractures of the specimens with different steel-slag-powder replacement ratios. In order to analyze the fracture morphology of specimens under different steel-slag-powder replacement ratio, an image-recognition method is utilized to divide the fracture morphology based on the different color-gamut ranges of coarse-aggregate fracture, coarse-aggregate surface, and cement-matrix fracture by setting the color-interval range from 30 to 230, as shown in Figure 3, where black is the coarse-aggregate fracture, blue is the coarse-aggregate surface, and the white part is the cement matrix. Based on this, in order to analyze the fracture performance of the specimen, the ratios of fracture area of coarse aggregate to the total fracture area of the specimen (referred to as coarse-aggregate fracture ratio R) under different steel-slag-powder replacement ratios are analyzed, as shown in Figure 4. As can be seen from Figure 4, within the range of steel-slag-powder replacement ratios set in this paper, coarse-aggregate fracture ratio R first increases and then decreases with the increase in steel-slag-powder replacement ratios, in which the R of SPC5 concrete reaches a peak percentage of 32.46%, which is 35.53% higher than the NC percentage of 23.95%. The concrete of specimens SPC10, SPC15, and SPC20 have 24.29%, 15.43%, and 8.03% of R, which are 1.42% higher and 35.57% and 66.47% lower than NC, respectively. Obviously, the larger the R is, the greater the fracture-energy dissipation of the specimens. This is mainly because the steel-slag powder contains hydration-activity substances such as calcium oxide and aluminium trioxide, which strengthen the bonding interface between the aggregate and cement paste to a certain extent. At a lower replacement ratio of SPC, the effect of steel-slag powder and steel-slag powder particles fill the concrete interior to improve the concrete-fracture strength [15], so the crack extension proceeds based on the lower fracture strength of the coarse aggregate. With the increase in steel-slag-powder replacement ratio, the fractures of the specimens are rougher and the area of the coarse-aggregate-cement matrix-debonding interface increases, while the coarse-aggregate fracture ratio R decreases. The reason may be that higher levels of steel-slag-powder replacement ratio lead to lower amounts of cement, which affects the overall cement material effect. In addition, too large a dose of steel-slag powder cannot improve the interfacial properties between the cement and the aggregate, producing more internal voids, so the strength will be reduced.

### 3.2. Fracture Energy

Fracture energy is the energy consumed to form a unit fracture surface, which is a major parameter reflecting the fracture performance and energy-consumption capacity of concrete. For the three-point bending notched beam, the fracture energy includes four parts, which can be expressed as
(1)$$W=W_{0}+W_{1}+W_{2}+W_{3}$$

$W_{0}$ is the work performed by the external force P, which is the envelope area under the load-deflection curve obtained in the test; $W_{1}$ is the work performed by the self-weight of the specimen before the external force P applied, which is small enough to be neglected compared with $W_{1}$; $W_{2}$ is the additional work performed by the self-weight of the specimen during the loading process [48], with $W_{2}=0.5\;mg\delta_{0}$. $W_{3}$ is calculated using the fracture-energy-calculation method proposed in the literature [49,50], after fitting a reasonable tail curve to the data measured by the test, The pertinent calculations are performed as suggested in Equations (2)–(5).
(2)$$P=\beta \delta^{-\lambda }\left(\beta ,\lambda >0\right)$$
(3)lnP=β−λlnδ
(4)$$W_{3}=\int_{\delta_{0}}^{\infty }\frac{\beta }{\delta^{\lambda }}d\delta =\frac{\beta }{\left(\lambda -1\right)\delta_{0}{}^{\left(\lambda -1\right)}}$$
(5)$$G_{F}=\left(W_{0}+W_{1}+W_{2}+W_{3}\right)/A_{lig}$$
(6)$$A_{lig}=t(h-a_{0})$$

The energy of each of the above components is calculated. The true fracture energy $G_{F}$, where $A_{lig}$ is the fracture area as shown in Equation (6). The concrete material at various mix ratios is obtained by Equation (5).

The load-deflection curves of the specimens are shown in Figure 5. The fracture energy of SPC is shown in Table 3. The fracture energy of specimens with different steel-slag-powder replacement ratios is shown in Figure 6. Compared with NC, it can be seen that the fracture energy of SPC5 and SPC10 concrete increases by 13.63% and 7.30%, respectively. The fracture energy of SPC5 is highest. With the increase in steel-slag-powder replacement ratios, the fracture energy of concrete gradually decreases. Therefore, SPC15 and SPC20, compared with NC, decreases by 2.04% and 9.62%, respectively. Steel-slag powder enhances the fracture energy of concrete for two reasons: On the one hand, it is related to the fracture-energy-absorptive capacity of concrete. During crack development, concrete needs to absorb a large amount of energy to destroy the internal cement-slurry-bonding force and aggregate shear stress. The hydration reaction of the active mineral of steel-slag powder can improve the fracture strength of the concrete matrix, effectively slowing the generation and propagation of cracks, and improving the energy-dissipation capacity of concrete. On the other hand, the steel-slag-powder particles enhance the compactness of the specimens. The addition of steel slag powder fills the cracks inside the concrete [51]. When the fracture surface expands further, the concrete needs to consume more energy to break. However, as the replacement ratio of steel-slag powder increases, the role of steel-slag powder becomes less obvious, which gradually reduces the strength of the concrete. At this time, the internal failure form of concrete changes the aggregate shear failure to interface peel failure, resulting in the gradual reduction in fracture energy.

During the propagation of concrete cracks, tensile failure of the coarse aggregate occurs, absorbing a large amount of energy, so there is a correlation between the fracture area of coarse aggregate and fracture energy. In order to analyze the correlation between fracture energy $G_{F}$ and the coarse-aggregate fracture ratio *R* and steel-slag-powder replacement ratio *C*, *C* and *R* are treated as independent variables in a regression analysis of the test data, finding the correlation coefficient r = 0.9987, through regression analysis to obtain the *C*, *R* regression equation as follows (7):(7)$$G_{F}=253.66+222.04C+492.65R$$

Equation (7) can better reflect the relationship between SPC fracture energy and steel-slag-powder replacement ratio, coarse-aggregate fracture ratio.

### 3.3. Fracture Toughness

The fracture toughness reflects the ability of the material to resist crack-instability propagation. In order to investigate the effect of different steel-slag-powder replacement ratio on the fracture toughness of SPC, the fracture toughness of the specimens in this paper is calculated and analyzed. For this paper, the fracture toughness $K_{IC}$ of the SPC material can be calculated according to Equation (8):(8)$$K_{IC}=\frac{P_{\max}}{th^{\frac{3}{2}}}f\left(\frac{a_{0}}{h}\right)$$
where $P_{\max}$ is the peak vertical load; h、t、s are the height, width, and span of the three-point bending specimen; $a_{0}$ is the depth of cut of the prefabricated specimen; $f(\frac{a_{0}}{h})$ is the geometric shape factor, whose expression is shown in Equation (9).
(9)$$f(\frac{a}{h})=2.9{\left(\frac{a}{h}\right)}^{1/2}-4.6{\left(\frac{a}{h}\right)}^{3/2}+21.8{\left(\frac{a}{h}\right)}^{5/2}-37.6{\left(\frac{a}{h}\right)}^{7/2}+38.7{\left(\frac{a}{h}\right)}^{9/2}$$

A fracture-process zone is formed at the front end of the precast crack in the concrete notched-beam specimen under load, while a critical crack is formed along the tip of the crack. The ASTM-recommended equation (Equation (8)) for fracture-toughness calculation is derived based on ordinary concrete materials. However, a new effective crack length $a_{c}$ needs to be used instead of a in Equation (8) for calculation of fracture toughness of SPC, and the same can be obtained using Equation (10) developed by Chaudhuri [52].
(10)$$a_{c}=\frac{2}{\pi }(h+h_{0})\arctan\sqrt{\frac{tE{\left(CMOD\right)}_{c}}{32.6P_{\max}}-0.1135}-h_{0}$$
where $h_{0}$ is the thickness of the thin steel plate at the blade of the device clamped extensometer (mm); $CMOD_{c}$ is the critical value of the crack-opening displacement (mm); E is the calculated modulus of elasticity (GPa). Among them, the calculated modulus of elasticity E of the concrete mix is calculated as follows:(11)$$E=\frac{1}{tc_{i}}\left[3.70+32.60\tan^{2}\left(\frac{\pi }{2}\frac{a_{0}+h_{0}}{h+h_{0}}\right)\right]$$
where $c_{i}=\frac{{\left(CMOD\right)}_{i}}{P_{i}}$ is the initial value of the specimen (mm/KN), calculated from the slope of the rising section of the specimen P-CMOD curve in Figure 7.

The load-crack-opening displacement curves (P-CMOD) of the five groups of specimens are shown in Figure 7. The details of calculations towards determining the fracture-toughness parameters and fracture-toughness values of SPC are shown in Table 4. The variations in fracture-toughness law of concrete mixes with different steel-slag-powder replacement ratios are shown in Figure 8.

It can be seen that the fracture toughness of the specimens increases and then decreases with the increase in the steel-slag powder-replacement ratio within the test range of this paper in Figure 8, which is consistent with the damage-mode pattern of the specimens. The fracture toughness of SPC5 increases by 53.22% as compared to NC, while the values for SPC10, SPC15 and SPC20 decrease by 0.31%, 17.65%, and 46.14%, respectively. It can be concluded that the appropriate amount of steel-slag powder can improve the fracture toughness of concrete. The reason can be illustrated as follows: On the one hand, the steel slag has cement properties to produce more calcium silicate hydrates in the concrete matrix with cement, which increases the bond strength of cement paste and aggregate. On the other hand, with smaller-sized particles, the addition of steel-slag powder optimizes the fine-aggregate gradation, and enhances the compactness of concrete and increases its mechanical strengths to different degrees. When the strength of the coarse aggregate is lower than the bond strength of concrete, cracks will propagate along the coarse aggregate, resulting in the concrete-crack propagation requiring more energy consumption. However, as the replacement ratio of steel-slag powder increases, the mechanical strengths begin to decrease, which leads to a continued decrease in the ability of concrete to resist the stable propagation of cracks. The main reason is that steel-slag powder cannot replace cement as the main cement material. Reducing the amount of cement makes the concrete bond strength decrease gradually. At this time, the failure form of concrete is changed from the shear failure of coarse aggregate to the failure of the concrete-bonding layer, causing the fracture energy to decrease. This indicates that an appropriate content of steel-slag powder improves the fracture performance of SPC.

### 3.4. Double K Fracture Model

In this paper, the fracture model describing the development of cracks in SPC using the double-fracture criterion is also established [53], which is calculated by the following Equation (12):(12)$$K_{IC}^{ini}=\frac{3P_{ini}S}{2h_{}^{2}t}\sqrt{a_{0}}F\left(\frac{a_{0}}{h}\right)$$
where $P_{ini}$ is the cracking load measured by the three-point bending test; S is the span between the beam supports in the three-point bending tests; *t* is the width of the notched-beam specimen (cm); h is the height of the notched-beam specimen; $a_{0}$ is the initial crack length (cm); $F\left(\frac{a_{0}}{h}\right)$ is the geometric shape factor calculated according to Equation (13). The unstable fracture toughness $K_{uni}$ of concrete is also calculated by the above two equations. The peak load $P_{\max}$ and the critical effective crack length $a_{c}$ are used instead of $P_{ini}$ and $a_{0}$ in the equation.
(13)$$F\left(\frac{a_{0}}{h}\right)=\frac{1.99-\left(\frac{a_{0}}{h}\right)\left(1-\frac{a_{0}}{h}\right)\left(2.15-3.93\frac{a_{0}}{h}+2.7{\left(\frac{a_{0}}{h}\right)}^{2}\right)}{\left(1+2\frac{a_{0}}{h}\right){\left(1-\frac{a_{0}}{h}\right)}^{\frac{3}{2}}}$$

Figure 9 shows the change law of double-fracture parameters of SPC under different steel-slag-powder replacement ratios of five groups of specimens, and the results of the double-fracture parameters of SPC specimens are shown in Table 5. The initial fracture toughness of the specimens decreases with the increasing steel-slag-powder replacement ratio, as observed in Table 5. Figure 9 shows the effect of increasing the steel-slag-powder replacement ratio on the calculated values of double-fracture parameters of SPC mixes. Among them, the initial fracture toughness of SPC20 is reduced by 41.22% compared with NC. There are two main reasons affecting the initial fracture toughness. On the one hand, the fracture strength of concrete depends on the concrete interfacial transition zone (ITZ) which is affected by the water-cement ratio and aggregate-cement matrix interface strength. On the other hand, when the modulus of elasticity of concrete increases, it is easier for a small deformation to produce initial cracks. The elastic modulus of concrete is increased by adding steel-slag powder appropriately. However, adding too much steel-slag powder will reduce the elastic modulus [54] and fracture toughness. In the process of fracture propagation of SPC with a low replacement ratio, the steel-slag powder can effectively prevent the rapid development and propagation of cracks. Until the unstable fracture toughness is reached, the crack will develop steadily when the load increases. The steel-slag powder has a positive effect on the unstable fracture toughness of concrete, as shown in Figure 9. It can be seen that the unstable fracture toughness of SPC5 increases by 46.79% compared with NC. With the increase in steel slag powder replacement ratio, the unstable fracture toughness of SPC decreases gradually, where SPC20 is 45.62% lower than NC. From above, the lower replacement ratio of steel-slag powder effectively improves the unstable fracture toughness of SPC. Moreover, it influences the unstable fracture toughness of the concrete with a high replacement ratio of steel-slag powder, which makes concrete beams more likely to enter into the unstable fracture-propagation phase. The reason is that the hydration reactions of steel-slag powder, along with those of cement, increase the strength of the cement matrix, resulting in the crack propagation along the aggregate cross section instead of the aggregate-cement-matrix interface, which takes more fracture energy. Therefore, the arrival of the crack-unstable fracture is delayed, which affects the speed of crack propagation. In addition, the reduction in cement is an important factor affecting the cementing property of the concrete. Adding too much steel-slag powder does not enhance the cementing property of the concrete. Cracks will rapidly propagate by passing aggregates, leading to the decline in the crack resistance of concrete, as similarly reported in a previous study [55]. Based on the results of the present set of experiments, the optimal fracture property of SPC is obtained when steel-slag powder replaces 5% of cement, when the cementing property can be significantly improved.

## 4. Conclusions

The influence of steel-slag-powder replacement ratio on the fracture performance of concrete is studied and concluded as follows:(1)The image-recognition method is used to analyze the fracture surface of the specimens, and the coarse-aggregate fracture ratio representing the ratios of fracture area of coarse aggregate to the total fracture area of the specimen is proposed to provide a quantitative-analysis method for the fracture-failure mode of the specimens. The results show that the coarse-aggregate fracture ratio R is consistent with the changing law of fracture-performance parameters under different steel-slag-powder replacement ratios which can imply the fracture mechanism.(2)Compared with concrete without steel-slag powder, a steel-slag-powder replacement ratio under 10% in weight can improve the fracture performance of the concrete, while a steel-slag-powder replacement ratio beyond 10% in weight shows a decreased effect, which implies positive and negative effects of steel-slag powder as a replacement to cement. The positive effect may be the filling effect of steel-slag powder, which strengthens the bond properties between the aggregate and matrix; and the negative effect may be the lower cementitious activity of steel slag compared with cement.(3)The fracture energy and fracture toughness of SPC first increases and then decreases with the increasing steel-slag-powder replacement ratio. The SPC with a 5% steel-slag-powder replacement ratio in weight shows the best fracture performance, while the fracture energy increases by 13.63%, fracture toughness increases by 53.22%, and compressive strength increases by 4.1% compared with NC, which may due to the addition of steel-slag powder strengthening the bond properties between the aggregate and matrix to a certain extent.

## Figures and Tables

**Figure 1 materials-15-02243-f001:**
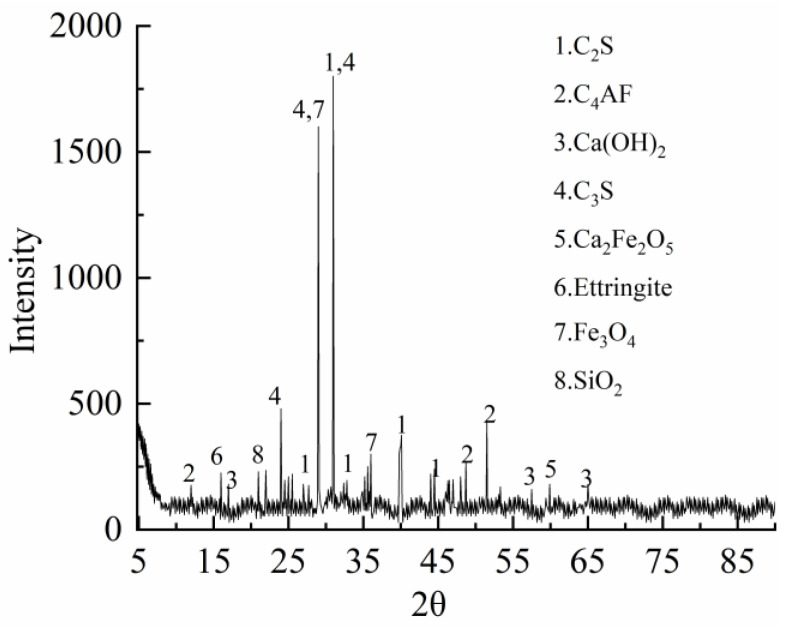
XRD pattern of steel-slag powder.

**Figure 2 materials-15-02243-f002:**
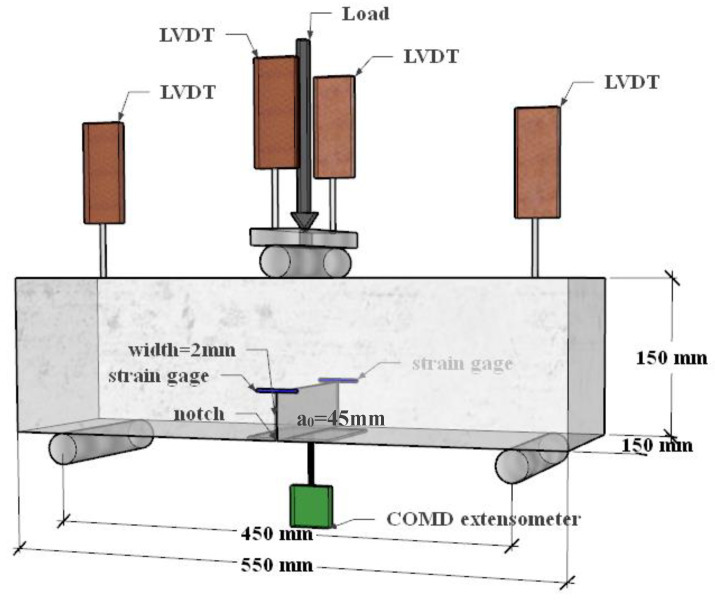
Experimental setup.

**Figure 3 materials-15-02243-f003:**
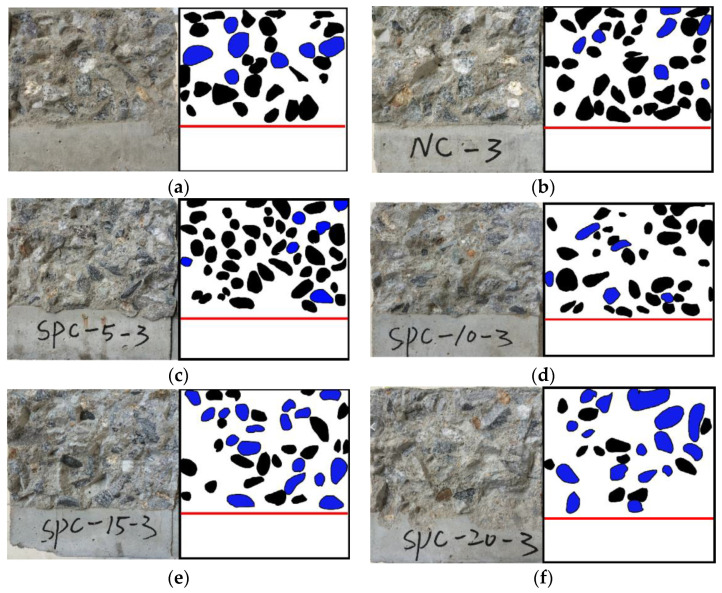
Fracture morphology of notched beam. (**a**) NC; (**b**) NC; (**c**) SPC5; (**d**) SPC10; (**e**) SPC15; (**f**) SPC20.

**Figure 4 materials-15-02243-f004:**
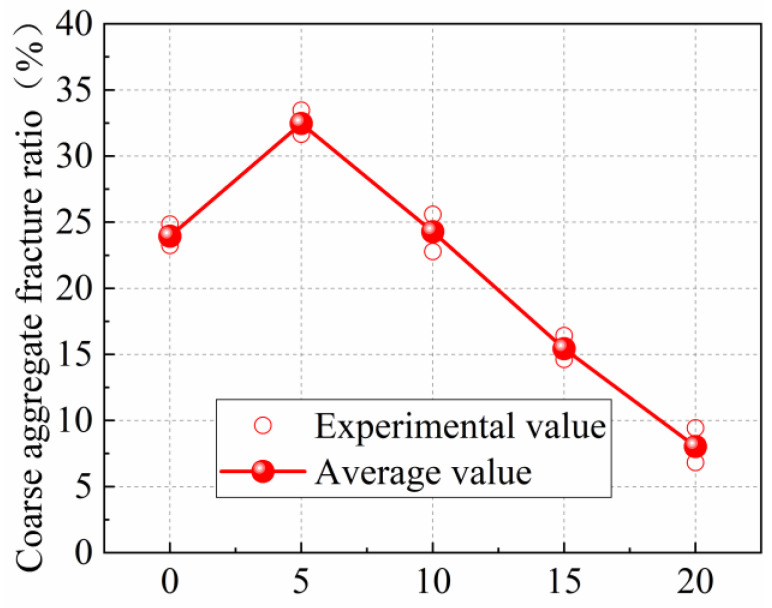
Relationship between coarse-aggregate fracture surface ratio R and steel-slag-powder replacement ratios.

**Figure 5 materials-15-02243-f005:**
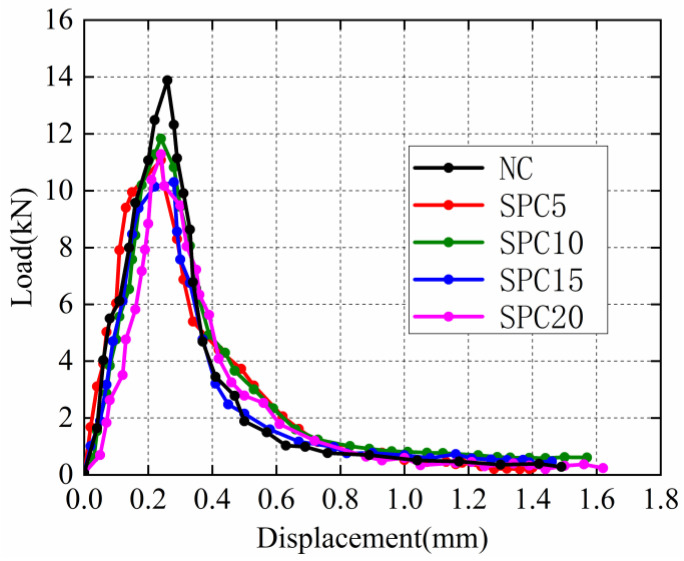
Load-deflection (P-δ) curves.

**Figure 6 materials-15-02243-f006:**
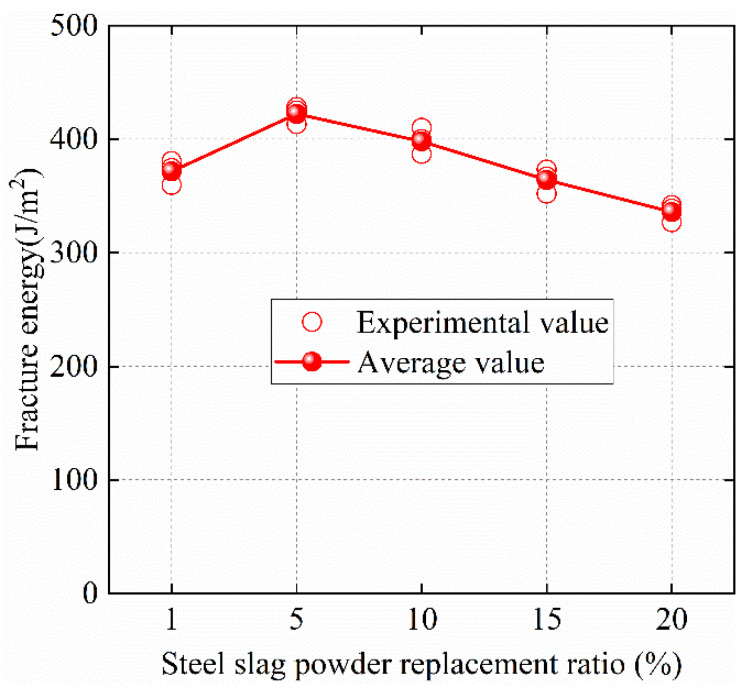
Fracture energy of SPC.

**Figure 7 materials-15-02243-f007:**
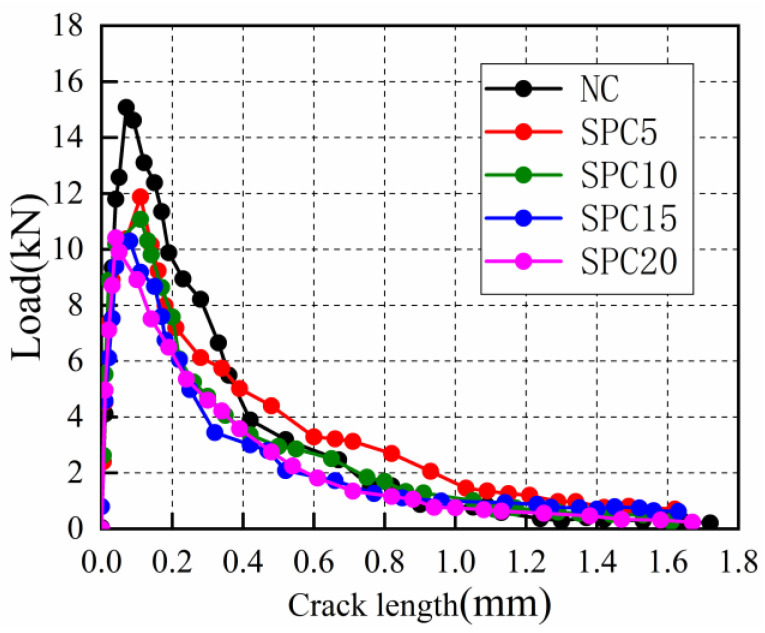
Load-crack-opening displacement curves (P-CMOD).

**Figure 8 materials-15-02243-f008:**
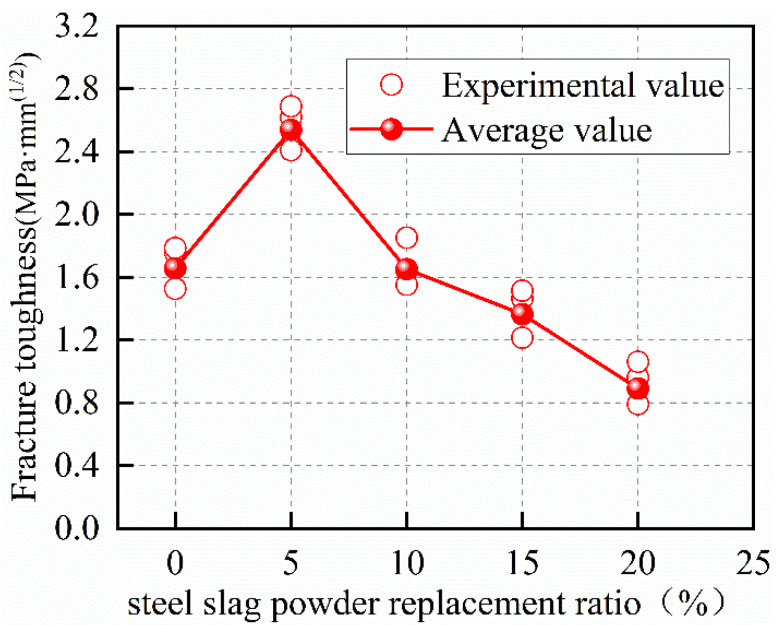
Fracture toughness of concrete at different steel-slag-powder replacement ratios.

**Figure 9 materials-15-02243-f009:**
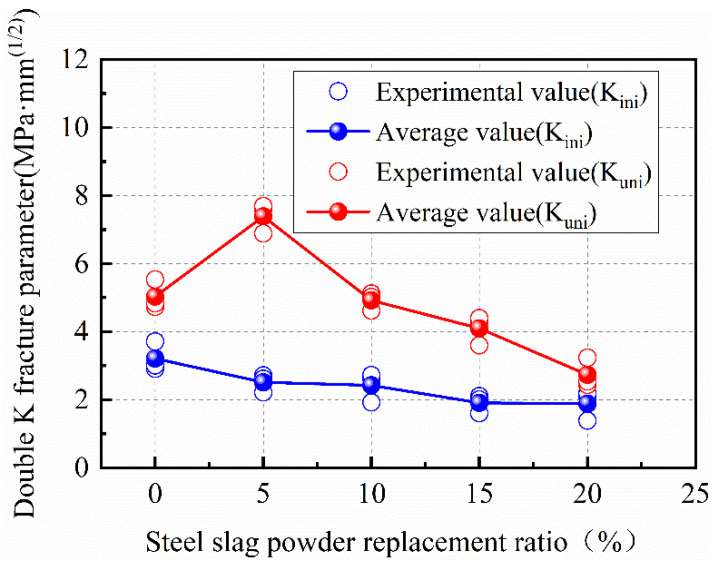
Double-K fracture parameters of concrete with different steel-slag-powder replacement ratios.

**Table 1 materials-15-02243-t001:** Chemical composition of raw materials (Mass fraction %).

Name	SiO_2_	Al_2_O_3_	Fe_2_O_3_	CaO	MgO	SO_3_	Na_2_O	K_2_O	R_2_O	Loss
Cement	25.44	7.06	2.89	55.32	2.25	2.77	0.44	0.67	0.88	2.28
Steel slag	15.42	4.45	26.79	38.48	8.08	0.14	0.17	0.12	0.25	6.10

**Table 2 materials-15-02243-t002:** Mix proportions of SPC (kg/m^3^) and the corresponding compressive strength (MPa).

Specimens	Water	Cement	River Sand	Coarse Aggregate	Steel-Slag Powder	Superplasticizer	Compressive Strength
NC	168.8	287.7	530.3	990.3	0	1.42	34.17
SPC5	168.8	273.3	530.3	990.3	14.4	1.42	35.57
SPC10	168.8	258.9	530.3	990.3	28.7	1.42	32.47
SPC15	168.8	244.6	530.3	990.3	43.1	1.42	29.25
SPC20	168.8	230.2	530.3	990.3	57.5	1.42	28.82

Note: Each group has 3 specimens. NC represents ordinary concrete; SPC represents steel-slag-powder concrete, with the number at the end of the specimens representing the replacement ratios of steel-slag powder with equal gravity.

**Table 3 materials-15-02243-t003:** Values of fracture energy of concrete of SFC.

Specimens	a_0_ (mm)	Mg (N)	δ_0_ (mm)	β	λ	R^2^	W_0_ (N·m)	W_2_ (N·m)	W_3_ (N·m)	A_lig_ (m^2^)	G_f_ (J/m^2^)	G_f_ (Average) (J/m^2^)
N C-1	46.5	287.6	0.37	0.4	2.44	0.9830	4.236	0.532	1.163	0.01552	382.140	371.600
N C-2	46.2	289.1						0.535			350.698	
N C-3	46.8	286.1						0.529			381.961	
SPC5-1	45.7	288.1	0.47	0.99	1.77	0.9695	4.17	0.677	2.299	0.01564	456.938	422.262
SPC5-2	45.5	289.9						0.681			353.150	
SPC5-3	45.4	286.5						0.673			456.698	
SPC10-1	45.2	287.6	0.47	0.85	1.90	0.9653	4.125	0.676	1.863	0.01572	423.930	398.744
SPC10-2	45.1	289.1						0.679			348.371	
SPC10-3	44.8	287.6						0.676			423.930	
SPC15-1	44.8	286.2	0.45	0.61	1.81	0.9878	3.662	0.644	1.438	0.01578	363.999	364.008
SPC15-2	44.7	288.0						0.648			364.255	
SPC15-3	44.4	284.6						0.640			363.770	
SPC20-1	45	285.2	0.47	0.59	2.25	0.9819	3.587	0.670	1.213	0.01575	347.308	335.861
SPC20-2	44.8	286.7						0.674			313.190	
SPC20-3	44.7	283.7						0.667			347.084	

**Table 4 materials-15-02243-t004:** Parameters and values of stress-intensity factor of the specimens.

Specimens	a_0_ (mm)	c_i_ (mm/KN)	h_0_ (mm)	mg (N)	Pmax (kN)	δ_0_ (mm)	CMOD_c_ (mm)	E (GPa)	A_c_ (mm)	K_IC_ (MPa·mm^(1/2)^)	K_IC_ (Average) (MPa·mm^(1/2)^)
N C-1	46.5	0.0033	1.45	287.6	15.08	0.26	0.07	26.86	61.90	1.663	1.66
N C-2	46.2	0.0034		289.1	15.24			24.88	59.75	1.612	
N C-3	46.8	0.0032		286.1	14.94			27.02	63.27	1.693	
SPC5-1	45.7	0.0026	1.45	288.1	11.88	0.24	0.11	33.13	90.79	2.565	2.54
SPC5-2	45.5	0.0027		289.9	12.04			30.55	88.81	2.465	
SPC5-3	45.4	0.0025		286.5	11.74			32.88	91.47	2.582	
SPC10-1	45.2	0.0031	1.45	287.6	11.08	0.24	0.08	27.29	77.55	1.711	1.65
SPC10-2	45.1	0.0033		286.1	11.24			24.64	74.40	1.614	
SPC10-3	44.8	0.0032		289.6	10.94			25.14	75.96	1.628	
SPC15-1	44.8	0.0041	1.45	286.2	10.30	0.28	0.08	20.34	70.11	1.345	1.36
SPC15-2	44.7	0.0038		287.7	10.21			21.09	72.40	1.402	
SPC15-3	44.4	0.0040		284.7	10.38			19.82	69.73	1.345	
SPC20-1	45.0	0.0038	1.45	285.2	10.41	0.24	0.04	22.11	48.73	0.898	0.89
SPC20-2	44.8	0.0037		283.7	10.24			21.74	49.69	0.899	
SPC20-3	44.7	0.0039		287.2	10.51			20.55	46.96	0.879	

**Table 5 materials-15-02243-t005:** The calculated results of double-K fracture parameters of SFC.

Test Piece	a_0_ (mm)	P_ini_ (kN)	A_c_ (mm)	Pmax (kN)	K_ini_ (MPa·mm^(1/2)^)	K_uni_ (MPa·mm^(1/2)^)	K_ini_ (Average) (MPa·mm^(1/2)^)	K_uni_ (Average) (MPa·mm^(1/2)^)
N C-1	46.5	12.67	61.9	15.08	3.229	5.056	3.21	5.04
N C-2	46.2	12.87	59.8	15.24	3.263	4.910		
N C-3	46.8	12.27	63.3	14.94	3.143	5.139		
SPC5-1	45.7	10.04	85.9	11.88	2.524	6.601	2.64	6.58
SPC5-2	45.5	11.03	84.8	12.04	2.764	6.523		
SPC5-3	45.4	10.56	86.4	11.74	2.641	6.611		
SPC10-1	45.2	9.76	77.5	11.08	2.433	5.088	2.43	4.92
SPC10-2	45.1	9.46	74.4	11.24	2.354	4.825		
SPC10-3	44.8	10.06	76.0	10.94	2.491	4.855		
SPC15-1	44.8	7.80	70.1	10.30	1.931	4.049	1.91	4.10
SPC15-2	44.7	8.00	72.4	10.21	1.977	4.205		
SPC15-3	44.4	7.40	69.7	10.38	1.820	4.049		
SPC20-1	45.0	7.62	48.7	10.41	1.893	2.757	1.89	2.74
SPC20-2	44.8	7.32	49.7	10.24	1.812	2.757		
SPC20-3	44.7	7.92	47.0	10.51	1.957	2.700		

## Data Availability

Not applicable.

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
