# Peer review of "Fracture Behavior of Steel Slag Powder-Cement-Based Concrete with Different Steel-Slag-Powder Replacement Ratios"

_materials, 2022, doi:10.3390/ma15062243_

Round 1

Reviewer 1 Report

The manuscript focused on the effect of different replacement ratios of steel slag powder as cement replacement on the fracture performance of concrete. The experiments are well-designed, and the obtained results are interesting and reliable. But some modifications are still needed before it can be published:

  1. The percentage replacement method of cement by steel slag powder is not mentioned in the abstract and methodology. Is cement and steel slag powder’s specific gravity are same? It is very important for a comparative study between two materials, so please add specific gravity values and justify if they are not the same.
  2. Please cite the paper correctly throughout the manuscript, e.g., check Lines 67, 72, 76, 87. Also, check many sentences started with a small letter; please fix them throughout the manuscript.
  3. The author needs to improve the introduction by addressing the work done in the past on the mechanical performance of cement-based materials containing steel slag powder as a replacement for cement. Subsequently, highlight the objectives of this paper that can fill up the gap on the lack of available research data and limited information regarding the studied properties of concrete containing steel slag powder.
  4. In Table 2, the purpose of using a superplasticizer is not clear in the body text as the cement content was relatively low (287.7 kg/m3) in the mix; please add more comments to understand the reader better. The slump test procedure, values, and discussion are required to add to the manuscript to see the impact. However, the methodology is not discussed in detail. What was the test age? How many specimens have been used for each mix?
  5. Correct the legend in Figer 9 and check others.
  6. Though the authors explained, indeed, a mechanism and in-deep analysis are needed to explain why the concrete containing 5% steel slag powder gives better fracture performance. Beyond this percentage, the fracture performance decreased. However, is their other mechanical strength properties, such as compressive and tensile strength, can be added to support the results.
  7. In conclusion, a 10% replacement ratio of steel slag powder was proposed, while it is not reported in abstract though mentioned that the SPC concrete showed best fracture performance with 5% steel slag powder replacement ratio. Add this proposal in the abstract as well.

Author Response

please see the attchment

Reviewer 2 Report

The article presents a large amount of experimental research. Methods are described adequately. The results are presented clearly, the conclusions fully substantiate the results obtained. However, authors are advised to check the article, as it contains a large number of technical typographical errors.

Round 2

Reviewer 1 Report

Thanks to the author for considering the comments. The authors significantly improved the revised manuscript by addressing the comments appropriately. Therefore, the paper can be accepted for publication.
